# Fatty Acids of Marine Mollusks: Impact of Diet, Bacterial Symbiosis and Biosynthetic Potential

**DOI:** 10.3390/biom9120857

**Published:** 2019-12-11

**Authors:** Natalia V. Zhukova

**Affiliations:** 1National Scientific Center of Marine Biology, Far East Branch of the Russian Academy of Sciences, Vladivostok 690041, Russia; nzhukova35@list.ru; Tel.: +7-423-231-0937; Fax: +7-423-231-0900; 2School of Biomedicine, Far Eastern Federal University, Vladivostok 690950, Russia

**Keywords:** fatty acids, mollusks, symbiotic bacteria, biosynthesis

## Abstract

The n-3 and n-6 polyunsaturated fatty acid (PUFA) families are essential for important physiological processes. Their major source are marine ecosystems. The fatty acids (FAs) from phytoplankton, which are the primary producer of organic matter and PUFAs, are transferred into consumers via food webs. Mollusk FAs have attracted the attention of researchers that has been driven by their critical roles in aquatic ecology and their importance as sources of essential PUFAs. The main objective of this review is to focus on the most important factors and causes determining the biodiversity of the mollusk FAs, with an emphasis on the key relationship of these FAs with the food spectrum and trophic preference. The marker FAs of trophic sources are also of particular interest. The discovery of new symbioses involving invertebrates and bacteria, which are responsible for nutrition of the host, deserves special attention. The present paper also highlights recent research into the molecular and biochemical mechanisms of PUFA biosynthesis in marine mollusks. The biosynthetic capacities of marine mollusks require a well-grounded evaluation.

## 1. Introduction

The importance of fatty acids (FAs) in marine environments commonly focus on polyunsaturated fatty acids (PUFAs), eicosapentaenoic acid (EPA, 20:5n-3), docosahexaenoic acid (DHA, 22:6n-3) and, to a lesser extent, arachidonic acid (ARA, 20:4n-6), which are vitally important not only to human health but also to health and survival of marine and terrestrial organisms. They are derived from two metabolically distinct n-3 and n-6 FA families. The metabolic precursor of EPA and DHA is α-linolenic acid (ALA, 18:3n-3), whereas linoleic acid (LA, 18:2n-6) is the metabolic precursor of ARA. It is common knowledge that animals and humans cannot synthesize both n-3 and n-6 PUFAs de novo. Nevertheless, they are required for normal development, growth and optimal health. They can be produced endogenously by humans, but the rate of their biosynthesis is too low to satisfy the physiological requirements. Thus, n-3 and n-6 PUFAs are considered as essential for important physiological processes and must be supplied in the diet. The beneficial effects of n-3 and n-6 PUFA supplementation in diets have been well established both for humans and for marine animals.

The major sources of n-3 PUFAs are aquatic food webs [1,2,3]. They play a key role in biological processes and are among the most important molecules transferred via the plant–animal interface in aquatic food webs. According to generally accepted views, PUFAs are produced de novo mainly by unicellular phytoplankton and seaweeds and further transferred from primary producers to consumers on the following trophic levels of the marine food chains [4]. The most physiologically important EPA and DHA are accumulated within aquatic ecosystems, as they are transferred to animals that can be consumed by humans. Numerous studies have shown the relationship of the FA composition of consumers and food consumed, and, therefore, FA can be used as efficient and useful biomarkers for the study of trophic interactions between organisms in aquatic ecosystems [5,6]. 

However, information about the endogenous mechanisms of marine invertebrates responsible for synthesis of n-3 and n-6 PUFAs is still being accumulated. Recent researches have shown the potential of some marine mollusks for endogenous synthesis of long chain PUFAs (LC-PUFAs) [7,8,9]. Based on the transcriptome and genome sequences, as well as various publicly available databases, a number of novel fatty acyl desaturases (*Fad*) and elongations of very long-chain fatty acid (*Elovl*) genes have been identified from the major orders of the phylum Mollusca, suggesting that many mollusks possess most of the required enzymes for the synthesis of long chain LC-PUFAs [10]. The question whether these findings of the desaturase sequences in invertebrate species really cast doubt on the idea that the organic matter is transferred along the food chains, and thus the existence of trophic links between primary producers and consumers and the relationship of the FA composition of animals and the FA composition of food, are currently under discussion [8].

Mollusk FA have attracted the attention of researchers that has been driven by their critical roles in aquatic ecology and in trophic food webs, as well as by their importance as sources of essential FAs with important impacts on human health [11]. Among marine animals, mollusks are especially important as a source of PUFAs (after fish). Many members of the phylum Mollusca, commonly known as clams and snails, are traditional seafood items in human diets, and rich in essential PUFAs. The edible mollusks are commercially harvested and cultured [12]. Marine bivalve mollusks are highly appreciated, partly because of their positive effects on human health arising from their constituents—highly valued n-3 LC-PUFA—and so their consumption is increasing every year [13]. The mollusks represent different trophic levels, trophic groups, and differ by various dietary habits. To date, extensive data has been accumulated on mollusk FAs. The great diversity of mollusks is accompanied by their wide chemodiversity because of their trophic preferences and defense modes, as well as the biosynthetic capacities that influence their chemical composition.

The main objective of this review is to focus on the most important factors and causes determining the biodiversity of the mollusk FAs, with an emphasis on the key relationship of these FAs with the trophic sources and the food spectrum, rather than to make a complete description of the FA composition of the known mollusk species. The marker FAs of the trophic sources are also of particular interest. The discovery of new symbioses involving invertebrates and bacteria, which are responsible for nutrition of the host, deserves special attention. The present paper also highlights recent research into the molecular and biochemical mechanisms of PUFA biosynthesis in marine mollusks. The biosynthetic capacities of marine mollusks require a well-grounded assessment.

## 2. Importance of Essential Polyunsaturated Fatty Acids for Human Health

FAs are involved in several biochemical pathways and, being an important source of energy and components of cell membranes, are responsible for determining their structure, functions and cell signaling [14]. They ensure fluidity of the lipid bilayer, selective permeability and flexibility of cellular membranes, and are responsible for the mobility and function of embedded proteins and membrane-associated enzymatic activities [15].

Many biological actions of PUFAs are mediated via bioactive lipid mediators produced by fatty acid oxygenases and serve as endogenous mediators of cell signaling and gene expression that regulate inflammatory and immune responses, platelet aggregation, blood pressure and neurotransmission [16]. They support the physiological functions as homeostatic mediator [17]. PUFAs n-6 and n-3 are precursors of signaling molecules with opposing effects. ARA is converted to prostaglandins, leukotrienes and lipoxins, whose effect is predominantly pro-inflammatory. In contrast, EPA- and DHA-derived eicosanoids have chiefly an anti-inflammatory effect. PUFAs n-3 exhibit the most potent anti-inflammatory effects that helps to control inflammation underlying many chronic diseases, including atherosclerosis, coronary heart disease, diabetes, rheumatoid arthritis, cancer and mental health [18]. A large number of epidemiological studies and clinical trials suggest a beneficial relationship between n-3 PUFA consumption and reduced inflammatory symptoms. So, EPA and DHA are capable of partly inhibiting inflammation reactions, including leukocyte chemotaxis, adhesion molecule expression, leucocyte–endothelial adhesive interactions, production of inflammatory cytokines, and T cell reactivity [19]. Low intake of dietary EPA and DHA is associated with increased inflammatory processes, general cardiovascular health and risk of the development of Alzheimer’s disease, as well as with poor fetal development, including neuronal, retinal and immune function [11,20,21]. Low maternal DHA intake may also cause increased risk of early preterm birth and asthma in children [22,23].

Many beneficial cardiovascular effects have been ascribed to PUFAs, including hypolipidemic, antithrombotic, antihypertensive, anti-inflammatory and antiarrhythmic properties, as well as the reduction of blood pressure [24]. The effectiveness of n-3 PUFAs for the prevention of cardiovascular diseases (CVD) is based on multiple molecular mechanisms, including membrane modification [25,26] where n-3 PUFAs are incorporated into lipid bilayer and affect membrane fluidity, formation of lipid micro-domains and also mechanisms such as attenuation of ion channels, regulation of pro-inflammatory gene expression and production of lipid mediators [27,28]. The use of n-3 PUFAs is recommended for ameliorating the CVD risk factors [11,29].

DHA, the dominant n-3 FA in the brain and retina, plays an important role in neural function, exhibits neuroprotective properties and represents a potential remedy against a variety of neurodegenerative and neurological disorders [30,31]. The potentially beneficial effect of DHA in preventing or ameliorating age-related cognitive decline has been revealed in a clinical study [30]. The n-3 LC-PUFAs exert positive effects on memory functions in healthy elderly adults [21] and support the neurological development of the infant brain [32,33]. Consumption of n-3 LC-PUFAs, particularly DHA, may enhance cognitive performance relating to learning, cognitive development, memory and rate of fulfilling cognitive tasks [34]. EPA and DHA play a critical role in neuronal cell functions and neurotransmission, as well as in inflammatory and immune reactions that are involved in neuropsychiatric disease states. Most experimental and epidemiological studies show the beneficial effect of n-3 PUFAs in various neurological and psychiatric disorders [35]. A diet supplemented with n-3 PUFAs exerts positive effects on brain structure and function in healthy elderly adults [36].

Several studies have confirmed that n-3 PUFAs possess a potential for prevention and therapy of several types of cancers and, moreover, they can improve the efficacy and tolerability of chemotherapy [37,38]. According to other studies, n-6 PUFAs, vice versa, induce progression in certain types of cancer [38]. Epidemiological and experimental studies have found a relationship between a PUFA-supplemented diet and the development of some types of cancer, including colon and colorectal carcinoma, breast cancer, prostate cancer, as well as lung cancer and neuroblastoma [38]. The promising effect of n-3 PUFAs on certain types of cancer is explained by their ability to modulate membrane-associated signal transductions and gene expression involved in cancer pathogenesis, as well as to suppress systemic inflammation [39].

Dietary intake of these essential components, as substances with therapeutic action, may maintain health, prevent the development of many diseases and mitigate a number of pathological conditions. Supplementation of PUFAs at a rate of at least 1 g per day, either in capsules or by marine products, demonstrated a protective effect against cardiovascular disorders, hyper-triglyceridemia, hyperlipidemia, metabolic syndrome or type 2 diabetes [29].

## 3. Primary Producers of Polyunsaturated Fatty Acids in Marine Ecosystems

### 3.1. Microalgae

Each algal class is characterized by a specific FA profile. The occurrence of certain compounds can be used as an FA signature for different algal classes. Chemotaxonomic differences in FA may be useful in the estimation of the input of specific microalgae in the tracing of these components on marine food webs.

Members of Bacillariophyceae are abundant in aquatic habitats and are considered as the most important primary producers of n-3 LC-PUFAs in marine food chains. Diatoms frequently dominate in seasonal phytoplankton blooms and, accordingly, these algae are the most studied classes of microalgae in terms of their lipids and FAs. The FAs reported for different species of Bacillariophyceae are typical for diatoms. The most abundant FAs are 20:5n-3 (it averages at 20–40% of total FA), 16:1n-7, 16:0, 14:0 and C16 PUFAs, 16:2n-4, 16:3n-4 and 16:4n-1, which account for about 80% of total FAs [40,41,42,43,44]. Hence, reliable markers of Bacillariophyceae have a high percentage of EPA, the predominance of 16:1n-7 over 16:0 and the presence of 16:2n-4, 16:3n-4 and 16:4n-1 along with low amounts of C18 PUFAs and DHA.

Dinophyceae species are major contributors to marine food webs and are second to diatoms as primary producers of organic matter in the oceans. They are especially abundant in coastal waters worldwide, where their exuberant growth, named algal bloom, is often observed. They are known as the main supplier of n-3 LC-PUFAs to marine animals. The more prominent FAs found in dinoflagellates are 16:0, 18:4n-3 (2.3–15.3% of total FA), 18:5n-3 (6.4–43.1%), 20:5n-3 (ranged from 1.8 to 20.9% in different species), and 22:6n-3, DHA (9.5–26.3%) [42,45,46,47]. Summing up the information on the FAs of this algal class, the high contents of 18:4n-3 and 22:6n-3 have generally been considered as useful signature compounds of dinoflagellates.

Green algae are classified into two classes, Chlorophyceae and Prasinophyceae, with their FA composition varying considerably. The most abundant FAs of the class Chlorophyceae are C18 and C16 PUFA isomers n-3 and n-6, of which, for example, 18:3n-3 reaches 43% of the total FAs [40,42,48]. The distinctive C16 PUFA isomers, 16:2n-6, 16:3n-3 and 16:4n-3, can be used in ecological studies as signature lipids to estimate abundance of these algae in phytoplankton or their input in the diet of invertebrates or transfer of these compounds into food webs. In general, the specific features of green algae are high concentrations of C16 PUFAs consisting of 16:2n-6, 16:3n-3 and 16:4n-3, and C18 PUFAs, such as 18:2n-6 and 18:3n-3, which are essential FAs and the precursors of metabolically distinct families of n-3 and n-6 PUFAs.

Eustigmatophyceae species contribute significantly to the organic matter of coastal waters in the Northern and Southern Hemispheres. Their FAs are dominated by three components, 16:0, 16:1n-7 and 20:5n-3, which together account for about 75% of the total FAs. In addition, an appreciable percentage of 20:4n-6 is detected (4–8.8%), whereas C18 PUFAs are present as minor components [42,49].

Cryptophyceae species are small marine or freshwater flagellates, which are abundant in some seasons and, hence, play an important role as food for invertebrates. A common characteristic of many cryptomonads is a very high proportion of n-3 PUFAs (up to 60–81.1% of total FAs) [42,50,51]. Among them, 18:4n-3 and 18:3n-3 are the most pronounced (together making up 40–50% of total FAs), but a high concentration of 20:5n-3 is also common (13–26%) [42,50,51]. Thus, the high percentage of 16:0, 18:4n-3, 18:3n-3 and 20:5n-3, along with a very low abundance of C16 PUFAs, is typical of most cryptomonads, which are considered as a highly valuable food source rich in n-3 PUFAs in aquatic ecosystems.

The class Prymnesiophyceae is divided into four orders, which have essential differences in lipid composition [42,52,53]. In general, the members of this class, similarly to diatoms, contain 14:0, 16:0, 16:1n-7 and 20:5n-3 as main components, but their distinguishing feature is the abundance of 18:4n-3 and 22:6n-3.

The FA profile of members of the Rhodophyceae is dominated by three major FAs, 16:0, 20:4n-6 and 20:5n-3, which together account for about 80% of the total FA [42,50]. It is worth noting that only red microalgae show a significant concentration of 20:4n-6 (up to 28%), which is a relatively rare or minor component in other classes.

Thus, the taxonomic differences in the FA composition between microalgae classes are obvious and each class is characterized by its specific FA profile. An FA analysis of microalgae has revealed signature compounds that may be useful to evaluate them as sources of different PUFAs. Uncommon FAs or groups of FAs may serve useful biochemical indicators in ecological studies. Chemotaxonomic differences, particularly those in terms of FAs, may be used for assessing the input of specific microalgae in the diet of animals.

### 3.2. Heterotrophic Protists

Another important source of PUFAs for marine mollusks is heterotrophic protists, zooflagellates and ciliates, constituting the links in the food web named the “microbial loop”. Among marine heterotrophic nanoplankton, flagellates are the dominant group in terms of abundance, biomass and diversity [54], while flagellates, in turn, are consumed by ciliates in the food chain. Heterotrophic protists, flagellates and ciliates, similarly to microalgae, are responsible for the production of LC-PUFAs, which are essential for organisms at higher trophic levels in marine ecosystems. The marine ciliate *Parauronema acutum* is reported to contain a significant level of PUFAs: 18:4n-3 (9% of total FAs), 20:5n-3 (10%) and 22:6n-3 (5%) [55]. A similar pattern exists for marine free-living heterotrophic flagellates [56]. The ability of zooflagellates and ciliates to efficiently produce n-3 PUFAs, 20:5n-3, 22:6n-3 and 20:4n-6, was proven experimentally [56,57]. Thus, zooflagellates and ciliates that constitute links of the microbial loop can be a source of PUFAs for suspension- and deposit-feeding mollusks in marine ecosystems [56,57,58].

## 4. Biochemical Markers for Identification of Mollusk Feeding Patterns

Due to the great structural diversity of FAs and their substantial taxonomic specificity, the identification of characteristic FA patterns at different trophic levels allows estimation of relationships between primary producers and consumers of different trophic levels of a food web [5,59]. The current trend in lipid biochemistry is the use of FAs as biochemical markers for determination of animals’ food sources and trophic relationships between species in aquatic communities [6,59,60,61]. The specificity of the FA composition of algae and microorganisms, which serve as food for consumers, are well documented (references for Section 3), and many of these FAs are transferred from prey to predators without modification [5,6,59]. This approach is based on the limited ability of animals to synthesize FAs, much of them animals receive from consumed food, particularly PUFAs, which can only be biosynthesized by microalgae and protozoa and become an essential dietary component for higher trophic levels. Potential food sources, such as diatoms, dinoflagellates, zooplankton and bacteria, have a distinctive FA composition with unique FAs or a specific FA ratio used as dietary tracers of mollusks (Table 1). For this reason, FAs are considered as biochemical markers, a very efficient and useful tool to provide information on the food spectrum and diversity of food sources for marine organisms and for studying food chains in marine ecosystems.

## 5. Fatty Acids of Marine Mollusks

Mollusks are extremely widely represented in the oceans, both in number of species and in density of populations. Of the seven classes of this phylum, Gastropoda, Bivalvia and Cephalopoda account for more than 95% of the mollusk species and are a major marine fishery resource. Plenty of information on the lipids and FAs of these classes, their commercial importance and, particularly, on their nutritional value as sources of n-3 PUFAs has been accumulated to date. In her review, Joseph emphasizes the important influence of environmental and biological factors on FA for members of this phylum [74]. Currently, new data are collected, which make it possible to review the features of the mollusks’ FAs and the impact of different factors on FA profiles. In this Section, we focus on the different diets of members of these classes, determining the principal differences in FAs between their species.

### 5.1. Gastropoda

The diet of gastropods, which are represented by the greatest number of species, differs according to the trophic group considered. According to the type of food, gastropods are generally divided into two groups: herbivorous and predators [75]. Their dietary specialization and trophic relationships are reflected in the FA composition of the species. Their trophic habits and food preferences influence the composition of their FAs, which can differ fundamentally for species with different diets (Figure 1).

Evidently, the most primitive type of gastropod feeding involves browsing and grazing of algae from rocks. High percentages of 16:1n-7 and 20:5n-3, typical of diatoms, have been found in the pelagic pteropod *Limacina helicina* that inhabits Arctic and Antarctic waters, indicating a strong evidence of diatom ingestion [78]. The limpet *Acmaea pallida* feeds most frequently on brown algae, while *Lottia dorsuosa* feeding on filamentous and unicellular algae, scraping them from the surface of stones. Consequently, FAs of algal origin found in snails, such as 18:3n-3, 18:4n-3, 20:4n-6 and 20:5n-3, reflects a herbivorous feeding strategy (Table 2) [76]. Two intertidal grazers, *Patella aspera* and *P. candei*, also exhibit high levels of EPA and ARA [79]. Meanwhile, in lipids of carnivores, *Cryptonatica janthostoma* and *Nucella heyseana*, 22:6n-3 is dominant, as a result of their animal diet (Table 2). These species are known as consumers of mollusks, mainly bivalves [75]. The FA composition of limpets and snails is characterized generally by predominance of 20:5n-3 and 22:6n-3, which constitute usually 25%–35% of total FAs, being a rich source of n-3 PUFAs.

In contrast, sea slugs dot not have this specific feature; these two marine PUFAs are minor components and in sum do not exceed 1–3% of total FA. FA profiles of nudibranchs differ principally from those of other mollusks in the abundance of numerous very long chain FAs (VLC FAs) specific for marine sponges [70,71,72] or by the high portion of tetracosapolyenoic acids (TCP FAs), produced by octocorals (Figure 1) [73]. The opisthobranchs, including sea slugs, are predators on sessile animals, such as sponges, corals, bryazoans and ascidians. The majority of nudibranchs are predators on sponges, and the occurrence of VLC FAs with double bonds at ∆5, 9, including 5,9–24:2, 5,9–25:2, 5,9–26:2 and *iso*-5,9–25:2, are certainly a result of feeding on sponges [77,80]. TCP FAs, 24:5n-6 and 24:6n-3, found in high proportions (each is more than 10% of total FAs) in the tritonid nudibranch *Tochuina tetraquetra*, originate undoubtedly from soft corals of the subclass Alcyonaria, which it feeds on [73]. FAs of the nudibranch *Armina maculate*, which feeds on a pennatulacean commonly named “Sea Pen”, *Veretillum cynomorium*, constituted predominantly 16:0, 18:0, 20:4n-6 and 20:5n-3 (62% of total FA); thus, evidencing a similarity with the FA profile of “Sea Pen” represented by the same major compounds, whereas FAs of the cephalaspidean *Aglaja tricolorata*, presumably feeding on foraminiferans from sandy bottoms, is rich in EPA and DHA (27% of total FAs) [81]. Moreover, the studied nudibranchs exhibit one more specific feature: their lipids are rich in n-6 PUFA, and their level is much higher than that of n-3 PUFAs. Dorid nudibranchs, besides 20:4n-6, contain also 22:4n-6 and 18:2n-6 [77]. High values of n-6 relative to n-3 PUFAs are unusual for marine organisms and are reported mostly for snails grazing on brown algae, being rich in 20:4n-6 (Table 1) [74,76].

### 5.2. Bivalvia

Most mollusks from the phylum Bivalvia are known to be suspension-feeders, their diet consisting mainly of plankton from the water column, protists from the near-bottom water layer and deposit-feeders collecting food from the surface of bottom sediments. Thus, planktonic and benthic microalgae, zooplankton, protozoans, including heterotrophic flagellates and ciliates, and also bacteria from detritus are the main components of diet of filter-feeding bivalves [75]. This feeding mode and, consequently, the diet primarily impact the composition of the mollusk FAs (Table 3), which exhibit an abundance of EPA, DHA, and quite often, ARA [61,74]. 

Variations in the trophic environment and also the food selectivity of the species result in the dominance of the FA biomarkers of diatoms or dinoflagellates, zooplankton or detritus, or a combination of these sources. The DHA to EPA ratio reflects the proportion of zooplankton, diatoms and dinoflagellates in the bivalve’s diet [6,82,83]. DHA often dominates in FAs of zooplankton and dinoflagellates [6,46,56,57], whereas EPA originates from diatoms [40,41,42]. A high concentration of 16:1n-7 and 20:5n-3, as well as a higher EPA/DHA ratio, suggests the importance of diatoms in the diet of the mollusks, whereas an elevated level of 18:2n-6, 20:4n-6 and DHA indicates the important contribution of microheterotrophs (flagellates and ciliates) in the diet. A higher proportion of odd-chain and branched FAs (OBFAs) is the evidence of the presence of bacteria in the diet of bivalves [84,85].

FA composition of the different taxa of marine bivalves from temperate waters of the East Pacific shows that their characteristic feature is a high abundance of n-3 PUFAs (Table 3). The concentration of both EPA and DHA reaches 25%, and ARA extends to about 8% of total FAs. FA composition varies from species to species, but n-3 PUFA are usually dominant. Furthermore, the high content of EPA and DHA shown in Table 3 is similar to the values obtained for the other species from different regions, for example, *Crassostrea angulata, Mytilus edulis, C. edule* and *Venerupis pullastra* from the coastal and estuarine systems of Portugal [86]; the oyster *Crassostrea virginica* [87] and sea scallop *Placopecten magellanicus* from the coast of Canada [88]; and the pod razor clam *Ensis siliqua* [89]. PUFAs, especially EPA (19–22% of total FAs) and DHA (20–32% of total FAs) were found to account for the majority of total FAs in tissues of the scallops *Patinopecten yessoensis* and *Chlamys farreri*, which provides an opportunity to use them as a potentially health-promoting food for human consumption [90]. Previous studies also reported the dominance of these PUFAs in tissues of *P. yessoensis* [84] and *Pecten maximus* [91]. 

In addition, spatio-temporal intraspecific variations in mollusk FAs are observed. So, FAs of *Pecten maximus* showed strong differences between individuals from shallow and deep-water habitats. This trend was driven by the content of marker FAs of diatoms, which are abundant near coasts. Scallops from deeper habitats are characterized by higher contents of flagellate FA markers compared with scallops from shallow habitats that emphasize the variability of the FA content according to the diet of this species along its distribution range [91]. FA biomarkers (Table 1) explain the spatial and temporal heterogeneity in nutrient sources for mollusks. The pattern of spatial and temporal variations of the biomarker FAs in the bivalve *Spondylus crassisquama* [83] and *Mytilus galloprovincialis* [92] revealed the nature and origins of food sources for these bivalves. Species-specific feeding adaptations to environmental variability of two bivalves, the clam *Callista chione* and the cockle *Glycymeris bimaculate*, from two shallow sites of the coastal oligotrophic Mediterranean Sea are revealed. The species demonstrate the differences in FAs mainly due to EPA and DHA percentage during the seasons. FA markers revealed a mixed diet where *Callista chione* fed more upon fresh material (diatoms and zooplankton) than *Glycymeris bimaculate*, which relied largely on bacteria-derived detritus [85].

*Pinna nobilis*, endemic to the Mediterranean Sea, is known to ingest different food items depending on its shell size. As a result, small-sized *P. nobilis* are associated with a detrital food chain characterize by saturated FAs (38%) and OBFAs (9.9%), while the diet of large- and medium-sized individuals have a greater proportion of PUFAs (EPA from 13% to 22% and DHA from 13 to 44% of total FAs). Thus, FA composition of the species reflects a lower contribution by markers of detritus and an increasing contribution of phytoplankton and zooplankton with increasing shell size [93].

### 5.3. Cephalopoda

Compared to data on lipids of gastropods and bivalves, information on cephalopods is not as abundant. Nevertheless, it is evident that their FA composition, similarly to that in gastropods and bivalves, is dietary dependent [94,95]. They inhabit pelagic ecosystems and are active predators preying on a variety of fish and invertebrates, such as crustaceans and mollusks. Their diet varies between species and is affected by gender, size, sexual maturity and season of year [96]. Cephalopods are generally known to be consumers of higher trophic levels, or top predators, actively accumulating n-3 PUFAs, EPA and, in particular, DHA in their tissues, which, are transferred up food chains from primary producers and ingested with their food [94]. They are excellent sources of n-3 PUFAs, especially EPA and DHA. An FA analysis of the most commonly consumed cephalopods, such as common cuttlefish *Sepia officinalis*, European squid *Loligo vulgaris*, common octopus *Octopus vulgaris* and musky octopus *Eledone moschata*, showed the dominance of DHA (21–39% of total FA), EPA (8–17%), ARA (1.5–12%), as well as saturated 16:0 (16–25%) and 18:0 (4–10%) during the seasons [97].

The FA composition of the mantle and digestive gland differed markedly between the squid species. The digestive gland is rich in monounsaturated FAs whereas the mantle contains high concentrations of PUFAs, particularly DHA (about 40% of total FAs) (Figure 2 and Figure 3). These findings imply that the squid, as a top predator, actively concentrates EPA and, in particular, DHA in the tissues from the diets. Published data show a similarity in FA of mantle tissue between various species from the different geographic regions (Figure 3), including *Nototodarus gouldi*, inhabiting the tropical and temperate waters of Australia and New Zealand [98], and *Moroteuthis ingens*, an endemic species to the Southern Ocean, having a circumpolar distribution in the sub-Antarctic [99]. Meanwhile, FAs of the digestive gland of squids differ significantly between species (Figure 3), largely reflecting the variety of diet consumed, e.g., [95]. An FA analysis, frequently applied in dietary studies of cephalopods, indicates that the digestive gland is an accurate source of dietary tracers [95], thus revealing a recent history of dietary intake [94,99].

A combination of stomach content and FA signature analyses provided clear evidence of seasonal shifts in prey composition of the arrow squid *Nototodarus gouldi* and suggested temporal variations in its diet. Additionally, FA analyses show dietary differences related with gender, size and maturity of females. According to these relationships, the diet of *N. gouldi* is closely associated with prey size, abundance, availability and, possibly, to life-history stages [98]. The spatial variations in diet are believed to result in the differences in FA profiles of the digestive gland of the onychoteuthid squid, *Moroteuthis ingens*, from four different areas of the Southern Ocean. The FA analysis indicates that crustaceans are an important prey for smaller squid, whereas fish constitute a major portion of prey of larger squid [94]. Moreover, on the example of the jumbo squid, *Dosidicus gigas*, it was shown that an FA analysis can trace the geographic origin of squid individuals [100].

The FA biomarker concept has proven to be useful in the study of energy sources for reproduction in the squid *Illex argentines*. It was found that the FA composition of ovaries shows a more pronounced correlation with that of digestive glands than with the mantle, an energy reserved organ that reflects the dietary intake. The similarity in FA composition between the ovaries and the mantle during the early maturation and spawning period indicates that during these two periods, the somatic energy reserve is involved in reproductive growth. Thus, the potential implication of FAs is useful to provide insights into the breeding strategies among cephalopods [101].

Octopus species consume mainly mollusks, crustaceans, fishes and, sometimes, smaller species of *Octopus* as supplementary dietary components. Significant differences between FAs of the tissues of common octopus *Octopus vulgaris* are evident [102]. Among PUFAs, ARA, EPA and DHA are present at high concentrations in all tissues. ARA is more abundant in the digestive gland compared to muscles (11.4 and 7.9%, respectively), DHA dominates muscle tissues (20.7 and 14.0%) and the percentage of EPA is similar in these tissues (about 15% of total FAs). In contrast to squids, monoenoic FAs are not the main components in octopus, but saturated FAs, 16:0 and 18:0, are prominent in all tissues. Furthermore, the muscles contain more 16:0 compared with the digestive gland (20.1 and 12.4%, respectively) and 18:0 content is similar in these tissues (about 12%) [102].

Cuttlefish is an inhabitant of the seafloor that ambushes small animals such as crabs, shrimps, fishes and small mollusks. Feeding experiments have demonstrated that the FA profile of the digestive gland of the cuttlefish, *Sepia officinalis*, reflects the FA profile of its prey. Cuttlefish that had been fed fish showed comparatively high levels of fish-derived signatures, and this dietary dependence was also found for cuttlefish fed on crustaceans [103]. The proportions of the specific prey FAs are mirrored on the animal FAs. The major FAs in the cuttlefish mantle are DHA, EPA, 16:0 and 18:0 [97].

Thus, cephalopods, being top predators, actively accumulate n-3 PUFAs, EPA and, in particular, DHA in their tissues, coming from their prey, and therefore are valuable marine sources containing high levels of DHA, EPA and a noticeable level of ARA.

## 6. Contribution of Symbionts to the Fatty Acid Pool of Mollusks

Symbiotic associations between mollusks and microorganisms are widespread; they result in unique ecological strategies and increased metabolic diversity of the partners (Table 4). Symbiotic microbes typically supply nutrients to host animals that provide the microbes with shelter.

Mollusks inhabit a variety of marine ecosystems. In environments characterized by poor nutrient contents, alternative strategies for nutrition have evolved. For example, some marine invertebrates, including mollusks living near deep-sea hydrothermal vents, cold seeps, on whales, wood falls on the deep-sea floor and shallow-water coastal sediments, derive their nutrition from chemoautotrophic microbes housed in their tissues and specialized structures [104]. The occurrence of symbiotic microbes with invertebrates that fix carbon dioxide autotrophically and synthesize organic compounds that are passed on to the host, play a critical role in establishing the lipid composition of the animals. Bacteria are known to produce various odd and branched FAs (OBFAs) named “bacterial acids” (Table 1). Additionally, *cis*-vaccenic acid, 18:1n-7, is biosynthesized by the anaerobic pathway unique for bacteria [111]. These FAs are widely offered as an indicator of the bacterial input in marine environment. Accordingly, elevated concentrations of the specific bacterial FAs, such as OBFAs, 16:1n-7 and 18:1n-7, coupled with a considerable reduction in n-3 and n-6 PUFAs produced by algae in lipids of the animals suggest a contribution of bacteria to the mollusk nutrition.

FAs have been used as a biomarker to reveal symbiotic relationships between bacteria and the bivalve mollusks *Solemya velum* [112], *Pillucina picidium* [113] and *Axinopsida orbiculata* from a shallow-water hydrothermal vent ecosystem of Kraternaya Bay [114], as well as the nudibranch *Dendrodoris nigra* [115]. These animals exhibit a high percentage of monoenoic FAs (about 40% of total FAs) mainly due to 18:1n-7, low concentrations of n-3 and n-6 PUFAs and an increased level of dienoic NMI FAs (Figure 4). In contrast, lipids of filter-feeding mollusks are dominated by 20:5n-3 and 22:6n-3, accounting for one-third of total FAs (Figure 4) [113]. A gastropod species, *Ifremeria nautilei*, from the deep-sea hydrothermal vent systems of the West Pacific, harbors two types of bacterial symbionts: a high abundance of sulphide-oxidizing and a low abundance of methane-oxidizing bacteria. It results in the dominance of 18:1n-7 (about 25% of the total FAs), 16:1n-7 (20–40%), and 16:0 (up to 15%) in its lipids and a low content of EPA and DHA [116].

Unlike bivalves from hydrothermal vents, deep-sea species living near cold seeps contain neither n-3 nor n-6 PUFAs. Two cold-seep bathymodiolin mussels, *Bathymodiolus japonicus* and *B. platifrons*, contain n-4 and n-7 PUFAs (25–27% of total FAs), including 18:3n-7,10,13; 18:4n-4,7,10,13; 20:3n-7,10,13; and 20:4n-4,7,10,13 with the main 16:1n-7 and 18:1n-7 (up to 25% in sum) [117], because they host methane-oxidizing bacteria and survive independently of photosynthetic products. A unique FA composition was reported for the cold-seep vesicomyid clam, *Calyptogena phaseoliformis*, which houses sulfur-oxidizing bacteria. The major FAs found in this clam belong to the novel n-4 and n-1 NMI PUFAs, such as 20:3n-4,7,15; 21:3n-4,7,16, and 20:4n-1,4,7,15, with significant levels of 20:2n-7,15 and 21:2n-7,16 as n-7 NMI FAs [118]. Similar traits exhibit another species of vesicomyid clams, *Phreagena* (synonym *Calyptogena*) *soyoae* and *Archivesica gigas*, harboring symbiotic sulphide-oxidizing chemoautotrophic bacteria in their gills. They are common in deep-sea chemosynthesis-based communities in the North Pacific. An FA analysis confirmed the lack of n-3 and n-6 PUFAs in their composition and revealed a high percentage of n-7, n-4 and n-1 NMI PUFAs. A comparison of FA compositions of various organs of the clams showed that the content of these NMI FAs in gills was much lower than that of other organs, it suggests that the biosynthesis of n-7, n-4 and n-1 NMI PUFAs occurs in tissues of vesicomyid clams [119].

Thus, in contrast to shallow-water filter-feeding or grazing mollusks, which contain photosynthetic n-3 and n-6 PUFAs as the main components, mollusks with symbiotic chemoautotrophic bacteria show significant modification of FA composition: A decline or complete lack of essential n-3 and n-6 PUFAs and appearance of significant amounts and a variety of NMI FAs, which can probably by synthesized by mollusks using the bacterial FAs as precursors.

The specific FA composition of symbiont-containing species may give a hint of the character of symbiosis. Photosynthetic symbionts, such as dinoflagellates from the genus *Symbiodinium*, settle on corals and giant clams *Tridacna* and supply photosynthetically fixed carbon to their hosts, which contribute to their lipid composition [120,121]. Some of these compounds, such as 18:4n-3, 18:3n-3, 18:5n-3, 22:6n-3 and 16:0, being biomarkers of symbiotic dinoflagellates [46,122], are detected in host organisms. Metabolic interactions consist in exchange of nutrients between the host and its symbionts [120], providing them with a competitive advantage in tropical waters poor in nutrients.

By comparing FAs of the herbivorous limpet *Acmea pallida* and nudibranch species, it is obvious that a striking feature of the nudibranchs is the unusually high level of OBFAs specific for bacteria (Figure 5). It is evident that the share of total OBFAs, predominantly 15:0, 17:0, 17:1n-8 and *iso*- and *anteiso*-C15, C16, C17, C18, and C19, in *A. pallida* is 0.7% of total FAs, whereas in the *D. nigra* it reaches 18.6%, in *Chromodoris* sp. 15.8% and in *Phyllidia coelestis* exceeds 30% [80]. They are normally minor metabolites in marine invertebrates, but the level of these bacterial acids recorded from nudibranchs proved to be extraordinary. A high level of bacterial FAs in nudibranchs may serve, in our opinion, as an indicator that the symbiotic bacteria provide the host with nutrients. Indeed, transmission electron microscopy (TEM) confirmed the presence of rod-shaped Gram-negative symbiotic bacteria in the cytoplasm of epithelial cells and the glycocalyx layer covering the epithelium of the notum and the mantle of *D. nigra* [115]. Moreover, some bacterial OBFAs, such as 17:1n-8 and 19:1n-8, evidently, serve as potential precursors for the biosynthesis of odd-chain PUFA, such as 21:2Δ7,13 identified in nudibranchs [80].

## 7. Biosynthesis of Fatty Acids in Mollusks 

Biosynthetic pathways of PUFAs are described in detail in many articles and reviews. In brief, monoenoic FAs, such as 18:1n-9 and 16:1n-7, are produced through the action of Δ9 fatty acyl desaturases. This activity is ubiquitous and found in all living organisms. Oleic acid, 18:1n-9, can be further desaturated through the action of n-6 or Δ12 desaturase to form, first, LA; then n-3 or Δ15 desaturase converts LA into ALA. Until recently it was believed that only plants are able to produce de novo LA and ALA, which are essential for animals [1,4,8]. The enzymes involved in LC-PUFA biosynthesis, namely, Δ8, Δ6, Δ5 and Δ4 desaturases, necessary for the production of EPA and DHA from 18:3n-3, have been described in algae [123]. 

Hitherto, there was a concept that the mollusks, as well as other marine invertebrates, are not able to synthesize n-3 and n-6 PUFAs de novo to satisfy physiological needs, and PUFAs in the marine invertebrates are derived exclusively from phyto- and zooplankton. This opinion was confirmed by experimental data on the incorporation of ^14^C-acetate in FAs of the yellow clam *Mesodesma macroides* [124,125]. Similarly, in the experiments with juvenile oysters, *C. gigas* fed algae that had previously been cultured with labeled ^14^C-acetate confirmed that dietary FAs are incorporated directly into oyster lipids, mostly in a unaltered form, and only less than 1% of the ^14^C-label was found in 20:5n-3 and 22:6n-3 [126]. Recently, it was shown experimentally that the majority of radioactivity from ^14^C-FAs incorporates into lipids of *Octopus vulgaris* paralarvae [127] and *Sepia officinalis* hatchlings [7], found as unmodified FAs with elongation being the only metabolism detected, and no desaturation activity towards the FAs was recorded. Moreover, in the study of the FA biosynthesis by bivalves, *Scapharca broughtoni*, *Callista brevisiphonata* and *M. edulis*, the active incorporation of ^14^C-acetate was found in saturated, monoenoic and dienoic NMI FAs, whereas the radioactive label in LC-PUFAs n-3 and n-6 families, as well as in their precursors, LA and ALA, was not detected. It has been found that NMI dienoic acids, 20:2Δ5, 11, 20:2Δ5, 13 and 22:2Δ7,13, 22:2Δ7,15, are the only PUFAs that mollusks are able to synthesize [128,129]. These FAs with isolated double bonds were suggested to be derived as a result of the action of Δ5desaturase and elongations. To produce these dienoic FAs, Δ5desaturase mediates the insertion of the double bond in 20:1n-7 and 20:1n-9, common for invertebrates [128]. Indeed, a fatty acyl desaturase (*Fad*) gene with Δ5 activity has been characterized both molecularly and functionally from the octopus *Octopus vulgaris* [102], the gastropod *Patella vulgata* [8], the abalone *Haliotis discus* [130], the noble scallop *Chlamys nobilis* [131] and the bivalve *Sinonovacula constricta* [132].

Thus, returning to the issue of the biosynthesis of FAs in mollusks, it should be notde that relying on the experimental data on radioactive acetate or FA incorporation it is proved that marine mollusks are not capable to synthesize de novo LA and ALA, and their long-chain homologues, EPA and DHA, which are essential acids and PUFAs, must be considered essential dietary nutrients.

With the development of genetic methods, interest in the issue of the capability of marine invertebrates to biosynthesize n-3 and n-6 PUFAs has risen and many noteworthy findings in this field have been made recently. The availability of gene sequences databases of multitude species of invertebrates contributes to understanding of the biochemical mechanisms of PUFA biosynthesis in marine invertebrates at a molecular level [8,9,133,134]. Using molecular genetics approaches, the *Fad* and elongation of very long-chain FA (*Elovl*) genes have been characterized both molecularly and functionally, namely, isolated, identified, and expressed in the yeast heterologous hosts. It has been proven that multiple invertebrates, including representative of mollusks, the gastropod *Patella vulgata*, possess the endogenous capability to produce n-3 PUFAs de novo and further biosynthesize physiologically essential n-3 LC-PUFAs [8]. Among aquatic invertebrates, the biosynthesis of LC-PUFAs has been more extensively investigated in marine mollusks [10,102,132,133]. Cloning and characterization of functional diversity of *Fad* and *Elovl* involved in the PUFA biosynthetic pathway was carried out in the cephalopods *Octopus vulgaris* and *Sepia officinalis*, abalone *Haliotis discus* and bivalves, *Chlamys nobilis* and *C. angulata*, and the achievements were reviewed [9]. The capability of LC-PUFA biosynthesis of particular species has been established to depend upon the complement of the key enzymes required, *Fad* and *Elovl* [9].

Thus, the presence of the *Fad* and *Elovl* genes coding the critical enzymes participating in the PUFA biosynthesis in invertebrates can be considered as the biosynthetic potential of mollusks to endogenously produce PUFAs.

## 8. Dietary Source of PUFAs Versus Own Biosynthetic Capability of Mollusks

The use of genetic methods contributed to a significant advance in the search of genes of the enzymes involved in the PUFA biosynthesis. Does the discovery of desaturase genes in a number of metazoans that enable them to endogenously produce PUFAs, actually mean that we should revise the concept of microalgae as primary producers of PUFAs and the subsequent transport of organic matter to invertebrates up the food chains and dietary origin of the PUFAs? It is worth clarification whether data on the presence of *Fad* and *Elovl* genes in mollusks can contradict the concept of the dietary dependence of fatty acids of animals on the fatty acid composition of their food.

The number of studied mollusk species, which possess genes encoding enzymes important in the LC-PUFA biosynthesis, is quite limited. Furthermore, the distribution of ωx desaturase genes within particular taxonomic groups is non-uniform, and the ability of the production of LC-PUFAs has been established to vary greatly between different mollusk species and heavily depend on the complementation of the desaturase and elongase genes, as well as on their enzymatic activity [9]. For example, ωx desaturase genes are found in the freshwater mussel *Elliptio complanate* and the common octopus *O. vulgaris* [102,134], but not in the marine oyster *C. gigas* nor the mussel *M. galloprovincialis* [8]. The presence of different types of biosynthetic enzymes, as well as their different enzymatic specificity, suggests that the abilities of mollusks for PUFA biosynthesis vary among species. A similar conclusion follows from the studies of biosynthetic capacity of mollusks using radio-labelled precursors [7,126,127,128,129].

The mechanism of gene expression is known to be complex and depend on various factors. Genes usually interact with and respond to the organism’s environment. *Fad* and *Elovl* genes are usually identified through a search for available sequence databases, and through an analysis of the distribution of ωx desaturase genes across the Phylum and functional characterization of the enzymes using a yeast heterologous expression system [8]. Marine ecosystems are highly rich in n-3 PUFAs produced by planktonic microalgae and, therefore, it can be assumed that the abundance of PUFA in the diet of mollusks may be among the environmental factors that can determine ωx desaturase gene expression. Probably, the genetically incorporated mechanism of PUFA n-3 synthesis is not implemented in mollusks, at least not to the full extent, since the amount of these vital components coming from food is sufficient for animals to provide their physiological and biochemical requirements.

However, in conditions of a PUFA deficiency in animals living in extreme environmental conditions, for example, in deep-sea ecosystems, there is no compensation for this deficiency by endogenous synthesis of PUFAs. As a result, a low content of PUFAs in mollusks or their absence is observed with the simultaneous compensatory increase in the number of NMI FAs [116,135,136,137,138,139]. There is ample evidence of the increase in NMI FAs with a shortage of PUFA [71,76,119,140]. A similar pattern is observed with the symbiosis of bivalves and chemoautotrophic bacteria, leading to a decrease in the PUFA and an increase in the NMI FA levels in the host [113,119,141]. This emphasizes the importance of understanding the biosynthetic capabilities of invertebrates, as well as the importance of combining dietary and biosynthetic approaches to understanding the origin of mollusk fatty acids.

Thus, some mollusk species, similarly to the majority of other invertebrates, possess genes encoding desaturases and elongases involved in pathways of biosynthetic pathways of LC-PUFAs [8,9,132,133], although both their potential and functions remain to be clarified. However, the most species of marine mollusks are apparently not capable to carry out the synthesis of these FA to a sufficient extent to satisfy the physiological requirements. The dependence of the FA composition of the mollusks on food consumed shows that n-3 PUFAs should be considered essential FAs, since their endogenous production appears to be limited.

## 9. Variations in Fatty Acids in Response to Environmental Factors

Numerous studies explore the influence of biotic and abiotic factors on their FA composition. The majority of studies focus on the seasonal fluctuations in the lipid and FA compositions that are found to directly relate with the reproductive cycle [142] or correlate with changes in the mollusk diet which followed the seston dynamics during the seasons [82,143]. Other studies assess the relationship between FAs and water temperature fluctuations [144,145]. Meanwhile, the species differ in their sensitivity to environmental factors. Since the importance of nutrient quality in terms of FA composition has already been addressed above, in this section the emphasis is on the environmental factors, such as bottom sediments, water salinity, temperature and water pollution. 

For benthic animals, the structure and composition of bottom sediments are the important characteristics of their habitat, being one of the key factors that determine the trophic potential of benthic epifauna. In muddy areas, a benthic community is composed mainly of diatoms, heterotrophic nanoflagellates and oligotrich ciliates, whereas in sandy areas, heterotrophic nanoflagellates, euglenoid flagellates, oligotrich ciliates and scuticociliates are dominant among protists [146]. For example, the FA compositions of the scallop *P. yessoensis* from two habitats, muddy and sandy, show pronounced differences pointing to different food availabilities. An elevated content of diatom biomarkers, 20:5n-3, 16:1n-7 and C16 PUFAs, indicates that scallops from a muddy bottom are more reliant on diatom sources. Scallops inhabiting a sandy site have higher amounts of biomarkers of flagellates, ciliates and invertebrate larvae, such as 18:2n-6, 20:4n-6 and 22:6n-3, compared to individuals from the muddy site. This suggests that a scallop’s diet depends on food availability [84].

Water salinity is one of the most influential environmental factors especially in estuarine systems and intertidal zone, where its variations cause major physiological and biochemical stress for aquatic organisms. Various species show different tolerance to water salinity. Under salinity stress, the bivalves *Cerastoderma edule* and *Scrobicularia plana* reduce food consumption and physiological pathways; a decrease in PUFA content is observed in *C. edule*. They can store the FAs which are of high physiological importance by reducing their activity and energy consumption [147]. The authors attribute the observed variations in the FA composition, particularly the contents of n-3 PUFAs, NMI FAs and n-6 PUFAs in the littoral mussel *M. edulis* from two different habitats, presumably to the necessity to survive the frequent fluctuations in such environmental factors as water temperature and salinity [148]. The observed modifications in the membrane lipids of the mussel gills lead to a change in the physical state of the membranes, their fluidity and permeability, the functions of ion channels, enzymes and receptors, which ensures the normal functioning of the organism under fluctuations in sea water salinity [148].

Intraspecific variations in FAs are also found to occur in response to water temperature variations. Negative relationship was observed between the acclimation temperature and the unsaturation index of membrane lipids in the oyster *C. virginica*, according to the homeoviscous adaptation theory. These temperature-related changes are mainly due to the variations in EPA content in fast-growing oysters, and in DHA and EPA contents in slow-growing animals [145]. The blue mussel *M. edulis* and the oyster *C. virginica* showed an increased 20:4n-6 level in their tissues as temperature rose, suggesting an increased availability of this FA for eicosanoid biosynthesis during stress response [144].

Contaminants, including the wide variety of pesticides and heavy metal pollutants, increase in marine ecosystems as the results of development of industrial and agricultural activities, being a stress factor for marine organisms. Such contaminants, like heavy metals, were reported to influence feeding, growth, reproduction, cardiac activity, and maturation of bivalve mollusks [149]. These physiological changes lead to changes in the lipid and FA metabolism, while the EPA level is reduced due to exposure to metals and organic pollutants [150,151]. A stress response, manifested as a decrease in PUFA and NMI FA content, was observed in the mussel *M. galloprovinciales* exposed to cadmium and copper [149], as well as to polycyclic aromatic hydrocarbon contamination [152]. A lower value of EPA compared to the control was recorded from the bivalves *Cerastoderma edule* and *Scrobicularia plana* exposed to copper sulphate [153], *Mizuhopecten yessoensis* exposed to cadmium [150] and *Scrobicularia plana* from a habitat contaminated by dioxin and polycyclic aromatic hydrocarbons [154], which can be considered a possible biochemical and physiological consequence for these animals.

Thus, the influence of some factors is not as evident; it is explained by the masking effect of a more significant contribution of food to the FA composition of the organisms. It should be noted that despite the variations in the FA composition of the filter-feeding mollusks in response to biotic and abiotic factors, their specific features, such as EPA and DHA dominance, are retained unchanged.

## 10. Conclusions

PUFAs, especially EPA and DHA, are fundamental to the health and survival of marine and terrestrial organisms. Mollusk FAs play critical roles in aquatic ecology and trophic food webs, and also play an important role as sources of essential PUFAs, significantly contributing to human health. This review highlights that the extensive taxonomic biodiversity of mollusks accompanies a wide chemical diversity, since the trophic preferences, biosynthetic abilities and physiological requirements of mollusks effect their chemical composition. The review provides evidence of a trophic transfer of FAs from various food sources to marine mollusks, which further emphasizes the nutritional contribution of the FA composition of animals. The variation in FA distribution and abundance between mollusks of different taxonomic and trophic groups is estimated. Mollusks differ in their feeding strategy, divided into the following trophic groups: filter-feeding, gathering, carnivorous, and the symbiont contribution. Some mollusks give shelter to microbes that provide them with nutrients, and these enable animals to settle in nutrient poor environments. Mollusks, which rely completely on bacterial endosymbionts in their diet, have specific FA profiles rich in “bacterial FAs” and poor in PUFAs. In addition, the biosynthetic potential of mollusks influences the FA composition. Mollusks are capable of producing numerous NMI FAs, which are especially important in low nutrient environments. Based on experimental data on incorporation of radioactive acetate or FA into marine mollusks, it is proved that they are not able to synthesize de novo LA and ALA and their long-chain homologues, EPA and DHA. However, it has recently been shown that some mollusk species possess genes encoding desaturases and elongases involved in pathways of biosynthetic pathways of LC-PUFAs. The biosynthetic capacities of marine mollusks require a well-grounded evaluation.

## Figures and Tables

**Figure 1 biomolecules-09-00857-f001:**
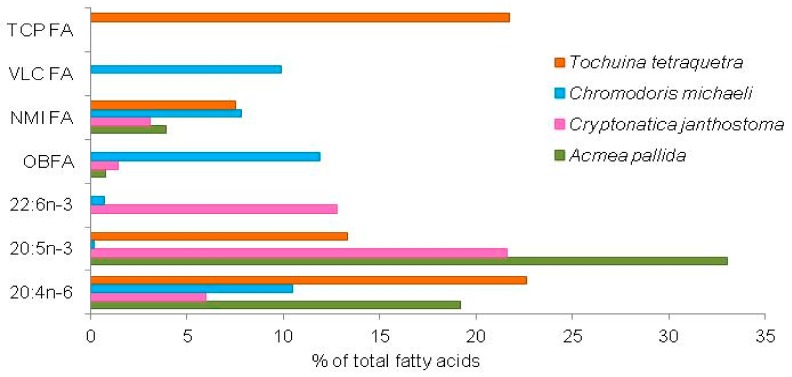
Distribution of fatty acids in gastropods with different types of feeding: herbivorous and carnivorous. Results are expressed as the mean [73,76,77]. TCP FA, tetracosapolyenoic fatty acid; VLC FA, very long chain fatty acid; NMI, non-methylene-interrupted; OBFA, odd-chain and branched fatty acids.

**Figure 2 biomolecules-09-00857-f002:**
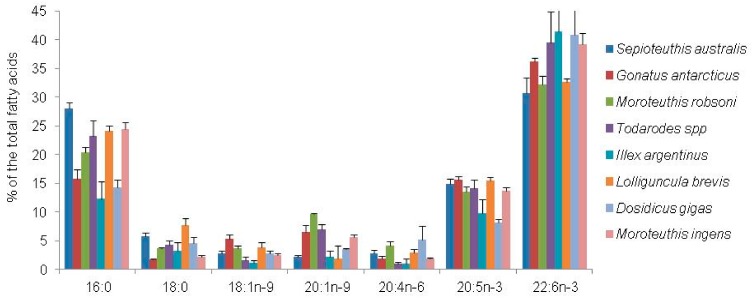
Major fatty acids (% of total FAs) in the mantle of squids. Values are mean ± standard deviation (SD) [94,95,99,100,101].

**Figure 3 biomolecules-09-00857-f003:**
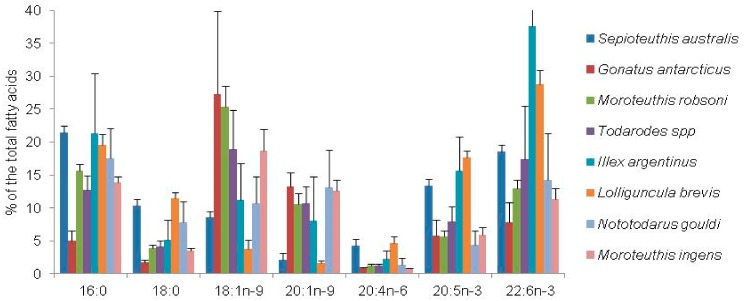
Major fatty acids (% of the total FAs) in the digestive gland of squids. Values are mean ± SD [94,95,98,99,101].

**Figure 4 biomolecules-09-00857-f004:**
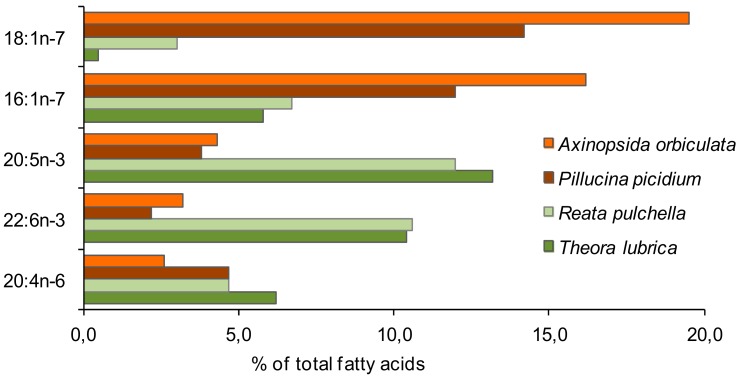
Distribution of the most remarkable of marker fatty acids of bacterial symbionts in the bivalve mollusks *Axinopsida orbiculata* [114], *Pillucina picidium* [113], containing sulfate-reducing symbiotic bacteria, and the symbiont-free bivalves, *Reata pulchaella* and *Theara lubrica* [113].

**Figure 5 biomolecules-09-00857-f005:**
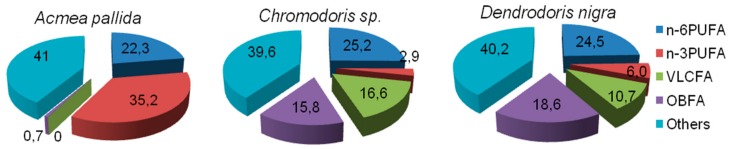
Distribution of fatty acids in the carnivorous nudibranch *Chromodoris* sp. [77], in *Dendrodoris nigra* [115] feeding on sponges and in the herbivorous limpet *Acmea pallida* [76] feeding on brown algae. *D. nigra* is known to harbor symbiotic intracellular bacteria [115].

**Table 1 biomolecules-09-00857-t001:** Fatty acids as biomarkers of food sources for mollusks.

Fatty Acid Markers	Food Source	References
20:5n-3, 16:1n-7/16:0 > 1, 14:0, 16:2n-4, 16:3n-4, 16:4n-1	Diatoms	[41,42]
18:4n-3, 22:6n-3	Dinoflagellates	[46]
18:2n-6, 20:4n-6, 22:6n-3	Heterotrophic flagellates	[56,57]
22:6n-3, 18:1n-9	Animal material Meiobenthos	[59,62]
15:0, 15:1, *iso*-15:0, *anteiso*-15:0, *iso*-16:0, 17:0, *iso*-17:0, *anteiso*-17:0	Heterotrophic bacteria	[63,64,65,66]
16:0, 18:0, 22:0	Detritus	[67,68]
18:1n-9, 18:2n-6, 18:3n-3, 18:4n-3, 20:4n-6, 20:5n-3	Brown algae	[69]
Very long-chain FAs: *iso*-5,9-25:2; 25:2Δ5,9; 26:2Δ5,9; 27:2Δ5,9; 26:3Δ5,9,19; 26:3Δ5,9,17; 27:3Δ5,9,19	Sponges	[70,71,72]
Tetracosapolyenoic acids: 24:5n-6, 24:6n-3	Soft corals	[73]

**Table 2 biomolecules-09-00857-t002:** Fatty acid composition of gastropod mollusks from the East Pacific (% of total FAs) [76].

Fatty Acids	*Acmea pallida*	*Lottia dorsuosa*	*Ischnochiton hakodadensis*	*Cryptonatica janthostoma*
14:0	0.4	4.6	4.5	3.4
15:0	0.3	1.0	0.5	0.6
16:0	5.9	13.9	13.2	6.4
16:1	0.8	5.8	3.8	2.8
17:0	–	0.5	–	–
16:3n-4	0.8	2.9	0.4	1.7
17:1n-8	0.4	–	0.4	0.8
18:0	6.3	4.5	–	8.8
18:1	13.2	15.9	15.7	3.6
18:2n-6	–	4.9	2.0	2.6
18:3n-6	–	–	0.5	0.2
18:3n-3	1.1	–	4.4	0.6
20:1	10.5	9.2	3.9	7.3
18:4n-3	1.1	2.2	1.7	0.5
20:2NMI	1.5	1.2	–	8.1
20:3n-6	1.6	0.9	3.5	4.2
20:4n-6	19.2	15.5	6.8	6.0
22:2NMI	3.9	4.2	3.6	3.1
20:5n-3	33.0	11.8	13.3	21.6
22:4n-6	–	0.7	4.2	1.1
22:5n-6	–	–	0.8	0.5
22:5n-3	–	0.9	4.4	2.6
22:6n-3	–	–	0.8	12.8

**Table 3 biomolecules-09-00857-t003:** Fatty acid composition of bivalve mollusks from the East Pacific (% of total FA) [76].

Fatty Acids	Arcidae			Mytilidae			Ostreidae	Cardiidae	Veneridae				Mactridae		Pectinidae	
	*Scapharca broughtoni*	*Arca boucardi*	*Anadara maculosa*	*Mytilus edulis*	*Crenomitilus grayanus*	*Modiolus difficilus*	*Crassostrea gigas*	*Clinocardium californiense*	*Callista* *brevisiphonata*	*Saxidomus purpuratus*	*Protothaca jedoensis*	*Mercenaria stimpsoni*	*Spisula voyi*	*Mactra chinensis*	*Patinopecten yessoensis*	*Chlamys swifti*
14:0	0.6	0.6	2.7	2.1	3.9	2.7	3.0	2.0	0.6	1.8	2.8	7.8	6.9	3.1	3.0	4.2
15:0	1.1	0.4	0.3	0.7	0.5	0.5	0.8	0.9	0.5	0.4	0.4	0.6	0.6	0.6	0.5	0.4
16:0	10.2	9.2	13.3	14.8	16.6	14.9	14.9	12.9	13.5	11.0	10.9	14.8	12.4	15.0	11.2	13.7
16:1n-7	1.5	3.2	2.4	5.0	8.3	5.5	4.9	4.9	2.4	3.3	6.6	9.2	5.6	9.6	5.0	4.9
16:3n-4	3.1	1.2	4.2	0.8	1.5	2.0	2.1	1.8	1.0	0.1	2.4	1.4	1.9	1.8	1.1	0.3
17:1n-8	2.2	1.3	0.2	0.2	0.6	1.4	1.0	2.4	2.0	0.6	0.8	0.6	0.8	0.8	08	0.9
18:0	10.6	6.6	13.5	3.5	2.8	5.8	3.6	5.5	0.5	5.8	4.0	6.4	6.8	4.0	6.1	5.3
18:1n-7	5.7	2.8	4.7	3.6	5.7	6.6	12.1	7.1	0.3	4.2	6.8	3.2	4.9	6.0	6.1	8.2
18:2n-6	2.2	1.6	3.8	1.7	2.2	2.1	2.2	1.2	0.5	0.8	1.3	1.4	1.3	0.4	9.0	1.6
18:3n-6	0.1	–	0.5	0.1	–	–	0.3	0.6	0.4	0.1	0.2	0.6	0.2	0.3	0.2	0.5
18:3n-3	1.0	2.1	1.5	1.4	1.5	1.5	1.8	1.5	0.2	0.7	0.3	0.7	0.8	0.4	0.4	0.9
20:1	10.8	12.5	8.9	12.3	5.6	7.4	6.8	3.2	12.5	17.3	5.8	3.6	6.5	6.8	3.9	5.5
18:4n-3	0.6	1.4	0.8	1.8	2.7	2.1	3.1	1.2	1.3	2.7	1.0	4.5	3.7	2.7	2.6	4.5
20:2NMI	0.1	0.7	0.1	0.8	3.9	1.2	1.3	3.4	0.2	0.3	0.2	1.9	1.0	–	0.9	0.2
20:3n-6	0.1	–		0.8	1.7	–	–	1.2	0.5	0.7	–	1.1	1.5	1.4	0.1	0.7
20:4n-6	5.8	6.4	7.8	3.9	2.5	3.3	2.1	4.3	3.6	2.9	3.5	1.5	2.7	2.1	3.0	3.8
22:2NMI	20.7	12.8	12.7	4.6	4.6	3.8	4.7	6.1	6.0	0.7	6.5	1.9	1.7	0.6	0.7	0.6
20:5n-3	6.1	10.3	4.0	14.5	16.2	22.9	16.7	13.4	18.3	22.3	14.4	24.5	17.3	21.2	19.7	20.2
22:3n-6	1.2	1.3	0.4	1.5	0.8	1.3	1.0	1.4	1.6	1.5	1.6	1.6	1.4	1.4	0.9	1.1
22:4n-6	0.9	0.8	1.3	0.1	0.2	0.4	0.1	2.5	1.1	2.2	1.4	0.4	0.8	0.8	0.1	0.2
22:5n-6	1.3	1.4	2.3	0.7	0.4	0.6	0.4	1.1	1.6	1.1	1.1	0.3	0.9	0.9	0.6	0.6
22:5n-3	1.0	1.3	0.7	1.1	0.8	1.1	0.1	3.0	2.0	3.3	1.5	0.1	2.0	2.0	0.5	0.9
22:6n-3	14.2	22.4	13.1	23.0	15.3	12.0	16.0	16.7	19.5	15.2	24.2	12.3	17.4	17.4	18.5	21.2

**Table 4 biomolecules-09-00857-t004:** Symbiotic microbes in marine mollusks.

Type of Nutrition	Symbionts	Function	Host	References
Chemotrophic	Bacteria	Nutritional	Bivalves and gastropods	[104]
Phototrophic	Zooxanthellae	Nutritional	Giant clam *Tridacna squamosa*, Gastropod *Strombus gigas*	[105]
	Algal chloroplasts	Nutritional	Sea slug *Elysia chlorotica*	[106]
	*Chlorella*	Nutritional	Clams, e.g., *Anodonta*	[107]
Heterotrophic	Bacteria	Nutritional	Bivalve shipworm *Bankia setacea*	[108]
		Light production	Squid *Euprymna scolopes*	[109]
		Chemical defense	Sacoglossan *Elysia rufescens*	[110]

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
