# Peer review of "Fatty Acids of Marine Mollusks: Impact of Diet, Bacterial Symbiosis and Biosynthetic Potential"

_biomolecules, 2019, doi:10.3390/biom9120857_

Round 1

Reviewer 1 Report

Title:  Fatty Acids of Marine Mollusks: Impact of Diet, Bacterial Symbiosis and Biosynthetic Potential

Author : Zhukova, N.V.

This paper of Zhukova wants to synthesize and discuss the scientific literature on factors and specificities of ecological niches determining the wide biodiversity of marine mollusk FAs.

I agree with the author that the source of essential PUFAs of the n-3 family that represents marine mollusk for human diet, and the now well-known beneficial health aspects of these FAs, is of interest for this review. But, in my opinion, this first large chapter reviewing the importance of n-3 LC-PUFAs for human health is far too long and comes out of the focus of this review (« Fatty Acids of Marine Mollusks: Impact of Diet, Bacterial Symbiosis and Biosynthetic Potential »). There are many papers, reviews on the subject and the author makes here a catalogue of the different and numerous beneficial effects on human health that is hard to read. I would try to modify this section trying to reduce it significantly in order to stick on the focus of the study, i.e. the specificities and diversity of marine mollusks FAs.

In the same way, the following chapter on FAs composition and specificities of primary producers in marine ecosystems (section 3) is very long and descriptive, which makes it hard to read. I believe this chapter could also significantly be reduced. The idea of a table like table 1 could help in this way.

In addition, I am not sure that section 4 on “FAs as biochemical markers for identification of mollusc feeding pattern” is relevant for this review. Indeed, many studies used and are using the important diversity of FA in marine primary producers and heterotrophic protists to try to estimate food sources and trophic ecology in filter feeders and mollusks, but once again, this would be the subject of another different review on this topic. In order to stick of the main aspect of the review, I suggest reducing and gathering these two paragraphs in order to be able to bring out the focus of this review (see above). In addition, the author frequently come back to this aspect of the relationship between feeding strategy/sources and FA composition of mollusks tissues in section 5. As an example, lines 226-231 conclude on the important diversity of FA compositions between microalgae that can be used for assessing input of specific microalgae in the diet of mollusks. I believe a more interesting aspect could be developed by putting emphasis on the fact that such biodiversity can be transfer to mollusks through their diet. Remember, as you mentioned and reviewed it thoroughly, that mollusks mainly accumulate EPA and DHA from their food sources (except for symbiotic associations). This biodiversity in the FA compositions of primary producers (and symbionts) can influence the composition of minoritary and other FA than EPA and DHA (or ARA), which can differ fundamentally within and between species with different diets and niches. I believe that these “minor” FA found in mollusks could represent a big interest both at the level of human diet but also for our understanding in the capacities of PUFA biosynthesis within the marine food web (as discussed and reviewed in this paper for mollusks).

This work presents an important and large review of FA in marine mollusks (section 5). As the author says, the main objective of this review is to focus on the most important factors determining the biodiversity of mollusk FAs, « rather than to make a complete description of the FA composition of the known mollusk species ». The fact is that section 5 is 7 pages long and quite unbalanced with sections 6, 7, 8 and 9. I think these sections 6, 7, 8 and 9 are the most interesting of this review, and would deserve to be further developed.

I believe that by reducing/rebalancing sections to the main question(s) of this work may help to clarify the conclusions and highlight the interests of studying and understanding the FA diversity in mollusks. I am not a native English speaker, but I think a revision of the English is also needed and would help to clarify and make reading more fluent for some sections of the manuscript.

See some specific comments below:

Line 36-37: Not needed. Repetition Line 72-79: It is a copy-paste of the last part of the abstract. Line 55-58: Not clear. Please check the wording in English Line 64-67: same Line 79: “The biosynthetic capacities of marine mollusks require a well-grounded evaluation”. Please check the wording in English. Section 3: At the end of each paragraph, author sum up the main information on remarkable FA composition of the different class described. This might be what author should only keep to reduce this chapter as suggested above.   Line 175: 18:4n-3 or 18:5n-3? Line 187-188: Please check the wording in English Line 209: delete repetition “ but among the studied species..” Line 215: please check wording (“attract the most attention”) Line 253-255: Repetition. Should be omitted Line 258: wording Legend Fig. 1: What are TCP FA? Table 2 and 3 is indicated 20:1. The n-x was not specified. I know that in your paper published in 1986 (ref 76 in the legend), this was not specified. But I believe since then you should have obtained other data. As mentioned and discussed in your original paper, they could be the biosynthetic precursors of NMI D, e.g. 20:1n-7 and 20:1n-9. But do you have an idea for 20:1n-11? Nota: Some of the data (bivalve species) of table 3 are not found in the reference 76 mentioned in the legend. Line 339-341: Indeed! See my comments above that mollusk mainly accumulate EPA and DHA from their food sources. Line 349-351: Repetition Line 402-404: Not necessary. It is repeating what is said just before in the paragraph Line 423-425 and 443-445: These concepts on 1- the interest of using digestive gland as an accurate organ for FA tracers and 2- the importance of lipids and FA during gametogenesis are indeed very interesting. First this is not only the case for cephalopods. If author wants to develop this idea, references and a discussion are missing for bivalves and gastropods. But once again, I ask the question if is it an objective of this review to discuss on the use of FA for trophic ecology in marine ecosystems. Line 463-465: Repetition. Not needed Line 495: Fig 4 does not show NMI levels. Line 558: Should be changed to : “For biosynthetic pathways, monoenic FAs, such as 18:1n-9……” Line 562: Author writes, “until recently it was believed that only plants are able to produce de novo LA and ALA”. Does the author means that it has been shown that other phyla have been demonstrated to possess and use the Δ12 and Δ15 for PUFA biosynthesis (e.g. metazoan)? If yes, this is an important point to specify here (see lines 615-618) Line 588-591: not necessary Line 594: omit “multitude” Line 605: what do you mean by “the achievements have been reviewed”? Line 606-607: this sentence is not clear. What the author wants to mean by “complement”? Line 618-621: please reword the sentence. Paragraph lines 643-651: Be careful with the use of PUFA; A NMI is a PUFA. We get lost in the idea developed here. I imagine author means EPA and DHA for PUFAs? What do you want to mean by “This emphasizes the importance of understanding the biosynthetic capabilities of invertebrates before attempting to underestimate the significance of dietary FAs »? Section 9: I think this chapter is the least accomplished although it deserves a real interest for the review to try to link such diversity of proportions of DHA, EPA and other FA in tissues of mollusks. Many studies have been conducted on these aspects and some important are missing that showed and discussed the roles of environmental factors on FA composition in mollusks. I think this chapter should represent a main aspect of the review. Line 662: Please omit “Because of the high commercial value of mollusks”. Line 667-670: not necessary. In addition, the paragraph that is following (line 671-682) deals only with trophic aspects, which is not the focus of this section. It should be omitted or moved elsewhere. Line 714-717 : contradicts what you say in this section?

Conclusion:

Line 725 : .. and physiological requirements Line 740: “The biosynthetic capacities of marine mollusks require a well-grounded evaluation “; What do you want to mean?

Author Response

I would like to express my sincere gratitude to the reviewer for the attention to my work and for the comments.

Comment: I agree with the author that the source of essential PUFAs of the n-3 family that represents marine mollusk for human diet, and the now well-known beneficial health aspects of these FAs, is of interest for this review. But, in my opinion, this first large chapter reviewing the importance of n-3 LC-PUFAs for human health is far too long and comes out of the focus of this review (« Fatty Acids of Marine Mollusks: Impact of Diet, Bacterial Symbiosis and Biosynthetic Potential »). There are many papers, reviews on the subject and the author makes here a catalogue of the different and numerous beneficial effects on human health that is hard to read. I would try to modify this section trying to reduce it significantly in order to stick on the focus of the study, i.e. the specificities and diversity of marine mollusks FAs.

Reply: The major interest in mollusk fatty acids has always been based on their value as seafood products rich in essential PUFAs, the physiological significance of which is shown in Chapter 2. This emphasis seems to me quite relevant, taking into account the scope of Biomolecules and the title of Special Issue “Fatty Acids in Natural Ecosystems and Human Nutrition”. In this section, which occupies slightly more than 1 page (out of 32), the main functions and roles of PUFAs in the organism are considered very briefly, almost schematically, and with no detail. This section is well structured. The first two paragraphs show the main effects of PUFA in the organisms: (1) their direct effect in membrane as acyl group of phospholipids and (2) as precursors of oxylipins. The following paragraphs summarize information on (1) beneficial cardiovascular effects, then (2) on the role DHA in neural functions, brain, retina is shown, while (3) the following paragraph describes n-3 PUFAs as a potential for prevention and therapy of several types of cancers; in conclusion, recommendation on dietary intake of n-3 PUFA are provided. Reducing this chapter means neglecting any of the most important roles, which is unreasoned in my opinion.

Comment: In the same way, the following chapter on FAs composition and specificities of primary producers in marine ecosystems (section 3) is very long and descriptive, which makes it hard to read. I believe this chapter could also significantly be reduced. The idea of a table like table 1 could help in this way.

Reply: Following the recommendation of the reviewer, I reduced the section 3.

Comment: In addition, I am not sure that section 4 on “FAs as biochemical markers for identification of mollusc feeding pattern” is relevant for this review. Indeed, many studies used and are using the important diversity of FA in marine primary producers and heterotrophic protists to try to estimate food sources and trophic ecology in filter feeders and mollusks, but once again, this would be the subject of another different review on this topic. In order to stick of the main aspect of the review, I suggest reducing and gathering these two paragraphs in order to be able to bring out the focus of this review (see above). In addition, the author frequently come back to this aspect of the relationship between feeding strategy/sources and FA composition of mollusks tissues in section 5. As an example, lines 226-231 conclude on the important diversity of FA compositions between microalgae that can be used for assessing input of specific microalgae in the diet of mollusks. I believe a more interesting aspect could be developed by putting emphasis on the fact that such biodiversity can be transfer to mollusks through their diet.

Reply: The section 4 consists of a table, which summarized the FA markers, and one paragraph of 6 sentences, which is a comment to the table. The reviewer fairly noted that I frequently refer to the data of this section, and, therefore, this section is extremely important for further explanation of the diversity of fatty acids in mollusks. It is, first, the diversity of the diet that leads to the diversity and variability of mollusk fatty acids. This is the concept that I present in the review.

Comment: This biodiversity in the FA compositions of primary producers (and symbionts) can influence the composition of minoritary and other FA than EPA and DHA (or ARA), which can differ fundamentally within and between species with different diets and niches. I believe that these “minor” FA found in mollusks could represent a big interest both at the level of human diet but also for our understanding in the capacities of PUFA biosynthesis within the marine food web (as discussed and reviewed in this paper for mollusks).

Reply: I fully agree with the reviewer as regards the potential importance of minor fatty acids. That is why I focused on the unusual and rare FAs identified in mollusks with symbiotic bacteria (Section 6). However, there are no publications available (or, at least, I do not know those) on their impact on human health, except for very rare works on the biological activity of columbinic acid and demospongic FAs.

Comment: This work presents an important and large review of FA in marine mollusks (section 5). As the author says, the main objective of this review is to focus on the most important factors determining the biodiversity of mollusk FAs, « rather than to make a complete description of the FA composition of the known mollusk species ». The fact is that section 5 is 7 pages long and quite unbalanced with sections 6, 7, 8 and 9. I think these sections 6, 7, 8 and 9 are the most interesting of this review, and would deserve to be further developed. I believe that by reducing/rebalancing sections to the main question(s) of this work may help to clarify the conclusions and highlight the interests of studying and understanding the FA diversity in mollusks. I am not a native English speaker, but I think a revision of the English is also needed and would help to clarify and make reading more fluent for some sections of the manuscript.

Reply: The section 5 is the main and central part of the review; it considers the diversity of mollusk FAs. It is divided into three subsections corresponding to the main mollusk classes. I see no need for rebalancing all sections in size.

English was revised by a native speaker.

Some specific comments:

Line 36-37: Not needed. Repetition

Reply: I do not consider it a repetition. The first sentence says that diet should be supplemented with PUFAs because they cannot be produced by humans, but the next sentence explains the health benefits of these fatty acids.

Line 72-79: It is a copy-paste of the last part of the abstract.

Reply: Traditionally, the goals and objectives of the study are placed at the end of the Introduction section, and, consequently, this part is naturally repeated in Abstract.

Line 55-58: Not clear. Please check the wording in English.

Reply: The wording is checked.

Line 64-67: same

Reply: The wording is checked.

Line 79: “The biosynthetic capacities of marine mollusks require a well-grounded evaluation”. Please check the wording in English.

Reply: The wording is checked.

Section 3: At the end of each paragraph, author sum up the main information on remarkable FA composition of the different class described. This might be what author should only keep to reduce this chapter as suggested above.

Reply: Done.

Line 175: 18:4n-3 or 18:5n-3?

Reply: Indeed, 18:4n-3 is especially important in dinoflagellates, and it is considered as a biomarker of these algae.

Line 187-188: Please check the wording in English

Reply: The wording is checked.

Line 209: delete repetition “but among the studied species..”

Reply: Done.

Line 215: please check wording (“attract the most attention”)

Reply: Checked.

Line 253-255: Repetition. Should be omitted

Reply: Repetition of what? Sorry, I could not find any repetition.

Line 258: wording Legend Fig. 1: What are TCP FA?

Reply: TCP FA, tetracosapolyenoic FA. This abbreviation is spelled out in the legend to Fig. 1.

Table 2 and 3 is indicated 20:1. The n-x was not specified. I know that in your paper published in 1986 (ref 76 in the legend), this was not specified. But I believe since then you should have obtained other data. As mentioned and discussed in your original paper, they could be the biosynthetic precursors of NMI D, e.g. 20:1n-7 and 20:1n-9. But do you have an idea for 20:1n-11? Nota: Some of the data (bivalve species) of table 3 are not found in the reference 76 mentioned in the legend.

Reply: To make Tables 2 and 3 more compact, some isomers are combined, including 20:1, which represents a sum of isomers, such as 20:1n-9 and 20:1n-7. A part of presented data on fatty acids of mollusks is my original data published earlier in my PhD thesis.

Line 339-341: Indeed! See my comments above that mollusk mainly accumulate EPA and DHA from their food sources.

Reply: Sorry, I did not understand your comment exactly. In this part, I indicate the sources of origin of FA in bivalves.

Line 349-351: Repetition

Reply: I could not find any repetition. What part?

Line 402-404: Not necessary. It is repeating what is said just before in the paragraph

Reply: The repetition that fish is the main food for cephalopods is deleted.

Line 423-425 and 443-445: These concepts on 1- the interest of using digestive gland as an accurate organ for FA tracers and 2- the importance of lipids and FA during gametogenesis are indeed very interesting. First this is not only the case for cephalopods. If author wants to develop this idea, references and a discussion are missing for bivalves and gastropods. But once again, I ask the question if is it an objective of this review to discuss on the use of FA for trophic ecology in marine ecosystems.

Reply: Yes, it is an objective of the review to demonstrate and prove that the diet, food spectrum, and food availability largely determine the fatty acid composition of mollusks.

Line 463-465: Repetition. Not needed

Reply: It is the conclusion from this subsection.

Line 495: Fig 4 does not show NMI levels.

Reply: In Fig. 4 only those fatty acids are shown which may serve markers of presence of bacterial symbionts in mollusks.

Line 558: Should be changed to : “For biosynthetic pathways, monoenic FAs, such as 18:1n-9”

Reply Thank you for your advice, but I would not like to begin the first and second sentences with the same phrase.

Line 562: Author writes, “until recently it was believed that only plants are able to produce de novo LA and ALA”. Does the author means that it has been shown that other phyla have been demonstrated to possess and use the Δ12 and Δ15 for PUFA biosynthesis (e.g. metazoan)? If yes, this is an important point to specify here (see lines 615-618)

Reply: “Until recently” means “until the discovery of desaturase genes in a number of metazoans” (Kabeya et al., 2018).

Line 588-591: not necessary Line 594: omit “multitude” Line 605: what do you mean by “the achievements have been reviewed”?

Reply: Corrected.

Line 606-607: this sentence is not clear. What the author wants to mean by “complement”?

Reply: complement = a set, a kit, not one but several

Line 618-621: please reword the sentence.

Reply: Done

Paragraph lines 643-651: Be careful with the use of PUFA; A NMI is a PUFA. We get lost in the idea developed here. I imagine author means EPA and DHA for PUFAs? What do you want to mean by “This emphasizes the importance of understanding the biosynthetic capabilities of invertebrates before attempting to underestimate the significance of dietary FAs »?

Reply: It is generally accepted that FA with regular double bonds in the chain, such as EPA and DHA etc., are traditionally referred to as PUFAs, whereas fatty acids with irregular double bonds, although they contain two or more double bonds, are referred to as NMI.

Section 9: I think this chapter is the least accomplished although it deserves a real interest for the review to try to link such diversity of proportions of DHA, EPA and other FA in tissues of mollusks. Many studies have been conducted on these aspects and some important are missing that showed and discussed the roles of environmental factors on FA composition in mollusks. I think this chapter should represent a main aspect of the review.

Reply: I could not ignore this topic and would like only to show that environmental factors modify the FA composition of mollusks. Indeed, there are many studies on this topic, and this can probably be a topic for another, separate review.

Line 662: Please omit “Because of the high commercial value of mollusks”.

Reply: Done

Line 667-670: not necessary. In addition, the paragraph that is following (line 671-682) deals only with trophic aspects, which is not the focus of this section. It should be omitted or moved elsewhere.

Line 714-717 : contradicts what you say in this section?

Reply: Changes in the environment (salinity, temperature, bottom sediments, etc.) naturally cause changes in food sources (composition of phytoplankton, zoobenthos, algae), which contribute to the fatty acid composition of mollusks. Thus, here I refer to the masking effect of food contribution.

Moreover, none of the environmental factors (salinity, temperature, pollution, etc.) can fundamentally changes the composition of fatty acids of mollusks, as it is observed in mollusks with different feeding strategies (see, for example, Fig. 1 in review).

Conclusion:

Line 725 : .. and physiological requirements

Reply: OK. It is added.

Line 740: “The biosynthetic capacities of marine mollusks require a well-grounded evaluation “; What do you want to mean?

Reply: Checked

Reviewer 2 Report

The manuscript by Natalia Zhukova provides the comprehensive information on the fatty acid biosynthesis  in the mussels

Author Response

I would like to cordially thank the reviewer for the high evaluation of my manuscript.

Reviewer 3 Report

Good review. I have enjoyed reading it. It is correctly structured and well written although the second half, particularly after section 7 needs formal edition. In fact, a full revision by a fluent English editor is advisable. Except for a few exceptions (marked in the text), the references are up to date. I have only a few comments. In terms of content, I think that the manuscript would benefit from a proper contextualization: what is the state of the art, whys is this type of review needed (beyond the beneficial effects of n-3 PUFA for humans). The first half of the work showing the effects of the trophic (fatty acid) markers on the body composition is quite solid. The last part of the work, integrating the biosynthetic information may not be as sound. It may be my perception, but I have the feeling that the author experiences the need to over justify that the influence of food on the final body composition is important. I do not think this is the case. In fact, the discussion around the biosynthetic capability and phenotypic-trophic influence is a very interesting point. I have added comments and suggestions in the pdf file. Please, consider to add a list of abbreviations. Note that the references in the reference list are not ordered, and there are other mistakes in this section: e.g. reference 123 is missing, and two references share 134 in the list.

Author Response

I would like to cordially thank the reviewer for the high evaluation of my manuscript, the attention to my work, for the comments, and suggestions provided in the pdf file.

Reply: English has been improved. References have been corrected. The references in the reference list are ordered. Comments and suggestions in the pdf file are taken into consideration in the revised version.

Explanations for some comments are below:

419. Please check this section. Reference to the figures is wrong. Mantle-tissue and digestive gland information is mixed up. N. gouldi: no mantle-tissue composition is presented.

Reply: I have checked this section carefully. References to figures 2 and 3 are OK.

Sepioteuthis australis [94]

Gonatus antarcticus [94

Moroteuthis robsoni [94]

Todarodes spp. [94]

Lolliguncula brevis [95]

Moroteuthis ingens [98]

Illex argentinus [100]

Dosidicus gigas [99]

Nototodarus gouldi [101]

As to Nototodarus gouldi, in the paper of Pethybridge et al., 2012 [101], data on fatty acids of this species is absent, whereas in Gong et al., [99] no data on digestive gland of Dosidicus gigas is provided, therefore, the references in these figures differ.

633-636. “Fad and Elovl genes were usually identified through the search of available sequence databases and analysis of the distribution of ωx desaturase genes across the Phylum; with the future investigation of the enzymes functions using a yeast heterologous system [8].”

Reply: According your recommendation this sentence is reworded as follows:

601-604 in revised version: “Fad and Elovl genes are usually identified through a search for available sequence databases, an analysis of the distribution of ωx desaturase genes across the Phylum, and functional characterization of the enzymes using a yeast heterologous expression system [8].” 649-651. “This emphasizes the importance of understanding the biosynthetic capabilities of invertebrates before 650 attempting to underestimate the significance of dietary FAs.” Who is trying to underestimate this dietary effect? This should be rewritten: both dietary and biosynthetic approaches are important for obtaining a “complete picture”

Reply: It is changed as follows:

617-620 in revised version: “This emphasizes the importance of understanding the biosynthetic capabilities of invertebrates, as well as the importance of combining dietary and biosynthetic approaches to identify the origin of mollusk fatty acids.”

692. May it be worth mo make a reference to membrane composition-fluidity?

Reply: It has been changed as follows:

660-663 in revised version. “The observed modifications in the membrane lipids of the mussel gills lead to a change in the physical state of the membranes, their fluidity and permeability, the functions of ion channels, enzymes and receptors, which ensures the normal functioning of the organism under fluctuations in sea water salinity [146].”

714-715. “Thus, the influence of some factors is not as evident; it is explained by the masking effect of a more significant contribution of food to the FA composition of the organisms.” I am not sure this sentence is clear:

When the influence of abiotic factors is not evident it may be explained by the stronger dietary influence?

Reply: Changes in the environment (salinity, temperature, bottom sediments, etc.) usually cause changes in food sources (changes in the composition of phytoplankton, zoobenthos, algae), which contribute to the fatty acid composition of mollusks. Thus, I here refer to “the masking effect of food contribution”.